# ComMutE-GAN: Comptetitive Multiple Efficient Generative Adversarial Networks

## Abstract

In complex creative scenarios, co-creativity by multiple agents offers great advantages. Each agent has a specific skill set and a set of abilities, which is generally not enough to perform a large and complex task single-handed. These kinds of tasks benefit substantially from collaboration. In deep learning applications, data generation is an example of such a complex, potentially multi-modal task. Previous Generative Adversarial Networks (GANs) focused on using a single generator to generate multi-modal datasets, which is sometimes known to face issues such as mode-collapse and failure to converge. Single generators also have to be very large so that they can generalize complex datasets, so this method can easily run into memory constraints. The current multi-generator based works such as MGAN, MMGAN, MADGAN and AdaGAN either require training a classifier online, the use of complex mixture models or sequentially adding generators, which is computationally complex. In this work, we present a simple, novel approach of training competitive multiple efficient GANs (ComMutE-GANs), with multiple generators and a single critic/discriminator, without introducing external complexities such as a classifier model. We introduce a new component to the generator loss during GAN training, based on the Total Variation Distance (TVD). Our method offers a robust, stable, memory efficient and easily parallelizable architecture. We present a proof-of-concept on the MNIST dataset, which has 10 modes of data. The individual generators learn to generate different digits from the distribution, and together learn to generate the whole distribution. We compare ComMutE-GANs with larger single-generator GANs and show its memory efficiency and increased accuracy.

## 1 Introduction

With respect to human beings, "Creators" refer to any and all who engage in creative thinking. When people learn about new topics, they create cognitive structures that allow them to understand the topics; they generate concepts that are new to them, although possibly already very well known to others. This is creativity at a strictly intra-personal level. When working in a social setting, such as a company or a classroom, one has to broaden this horizon to include "Co-creativity"."Creativity through collaboration" summarizes the definition of co-creativity as defined by Lubart (2017). Often, collaborators have different or complementary skills that enable them to frequently produce shared creations that they could not or would not produce on their own (Lubart & Thornhill-Miller, 2020). AI-aided co-creation has also been proven to improve general well-being (Yu et al., 2021).

Generative Adversarial Nets (GANs) are implicit generative models where one or more generators play a zero-sum game with a discriminator to recreate and potentially expand a chosen dataset. According to the definition established above, the generator models in modern day GANs such as those described in the works of Karras et al. (2018); Sauer et al. (2022); Karras et al. (2020); Goodfellow et al. (2014); Radford et al. (2015); Arjovsky et al. (2017); Gulrajani et al. (2017) exhibit creativity on an intra-personal level. Accordingly, generative networks have been applied in many creative applications such as painting (Ganin et al., 2018; Mellor et al., 2019; Parikh & Zitnick, 2020), doodling (Ha & Eck, 2017; Cao et al., 2019) and extending fictional languages (Zacharias et al., 2022). Most noticeable, in all the applications listed above, a single, large, generative agent was applied to perform a complex task rather than breaking it down into smaller, more easily manageable sub-tasks. This approach, although effective, is upper-bounded by memory constraints. Inspiration

from co-creativity research aims to resolve these constraints.

Other implementations of GANs that use collaboration, such as MGAN (Hoang et al., 2017), MM-GAN (Pandeva & Schubert, 2019), MADGAN (Ghosh et al., 2018) and AdaGAN (Tolstikhin et al., 2017) try to rectify the missing co-creativity functionality of GANs by using a mixture of multiple generators, modeling the input latent space as a mixture model, requiring the discriminator to classify as well and sequentially train and add generators to the mixture, respectively. MGAN and MADGAN require the discriminator to have classification capabilities and both implementations force the generators to generate different modes. The AdaGAN implementation poses a problem of computational complexity because of the sequential training nature. MMGAN on the other hand focuses on using mixtures at the input latent space level rather than on separating generators.

Our work stands to provide an easier approach to co-creativity than the ones presented above. It does not require the online training of a classifier, we do use a pre-trained MNIST classifier to augment the generator loss with the Total Variation Distance (TVD) to enforce mode separation during training, but mode separation maybe enforced using other methods, not requiring this pre-trained classifier at all. We also show that this is easily and efficiently scalable to more than two generators, which can all be trained in parallel without making complex transformations to the latent space.

Analogous to human behaviour in a social setting, collaboration among two or more such generators allows each individual generator to focus on and specialize in a specific sub-task, making the training process more stable, more efficient (memory-wise), the generated images clearer and the distribution of the generated images closer to the actual distribution of the chosen MNIST dataset.

## 2 BACKGROUND

### 2.1 GANS, WGANS AND WGAN-GP

Generative Adversarial Nets (GANs) (Goodfellow et al., 2014) consist of two neural networks: a *generator* ($G(\mathbf{z})$) that takes in randomly sampled noise, and generates fake images; a *discriminator* ($D(\mathbf{x})$), that takes in batches of real and fake data points and outputs a 0 for fake images or 1 for real ones. They minimize the well-known GAN objective function

$$\min_G \max_D \mathbb{E}_{\mathbf{x} \sim P_r}[log(D(\mathbf{x})] + \mathbb{E}_{\tilde{\mathbf{x}} \sim P_g}[log(1 - D(\tilde{\mathbf{x}})] \tag{1}$$

Where $\tilde{\mathbf{x}} = G(\mathbf{z})$, $\mathbf{z} \sim P_z$, $P_r$ is the data distribution, $P_g$ is the generator distribution and $G$ and $D$ are the *generator* function and *discriminator* function, respectively.

A variation of this method is the Deep Convolution GAN (DCGAN) (Radford et al., 2015), that uses Convolutional Neural Networks (CNNs), to improve image data generation significantly.

Wasserstein GAN (WGAN) (Arjovsky et al., 2017) improves vanilla GANs by presenting and rectifying problems with the original GAN objective function. In WGANs, the Wasserstein distance between real data and generated data is minimized by optimizing over the objective

$$\min_G \max_{D \in \mathbb{D}} \mathbb{E}_{\mathbf{x} \sim P_r}[D(\mathbf{x})] - \mathbb{E}_{\tilde{\mathbf{x}} \sim P_g}[D(\tilde{\mathbf{x}})], \tag{2}$$

The WGAN-GP method (Gulrajani et al., 2017) is an improvement over this method which uses their novel *gradient penalty* loss, instead of the former *weight clipping* to implicitly enforce the Lipschitz constraints. The new objective function solved by WGAN-GP is given by

$$\min_G \max_{D \in \mathbb{D}} \mathbb{E}_{\mathbf{x} \sim P_r}[D(\mathbf{x})] - \mathbb{E}_{\tilde{\mathbf{x}} \sim P_g}[D(\tilde{\mathbf{x}})] + \lambda * \mathbb{E}_{\hat{\mathbf{x}} \sim P_{\hat{\mathbf{x}}}}[(||\nabla_{\hat{\mathbf{x}}} D(\hat{\mathbf{x}})||_2 - 1)^2], \tag{3}$$

where the first two terms are the same as equation (2) and the third one is the *gradient penalty* (GP) term. Here, $\hat{\mathbf{x}} = \epsilon \mathbf{x} + (1 - \epsilon)\tilde{\mathbf{x}}$ and $\epsilon \sim U[0, 1]$. $P_{\hat{\mathbf{x}}}$ is the probability distribution associated with $\hat{\mathbf{x}}$. Modern GANs such as Karras et al. (2018); Sauer et al. (2022); Karras et al. (2020) that provide state-of-the-art performance on large-scale image synthesis improve these basic methods by making their models larger and training procedures more complex. Our method explores another direction, namely, more instances of compact generators competing with each other. The idea takes inspiration from social dynamics and co-creativity research.

## 2.2 GANS FOR MULTI-MODAL DATASETS

In past works, there have been attempts with varied scope, to learn multi-modal datasets such as multi-class generation. The most notable are the MMGAN by Pandeva & Schubert (2019), MGAN by Hoang et al. (2017), MAD-GAN by Ghosh et al. (2018) and AdaGAN by Tolstikhin et al. (2017). MMGAN approaches multi-modal generation task by considering the latent space **z** as a Gaussian mixture model consisting multiple modes. Initially, a Gaussian distribution is sampled from a cluster of Gaussian distributions ($Cat(K, 1/K)$), where the dataset has K "clusters". Then, **z** is sampled from this distribution and transformed by the generator to create images. This method uses an extra *encoder* (E) model to predict the cluster of data objects.

The MGAN method (Hoang et al., 2017) implements multiple generators that generate $K$ different batches. One of these batches is sampled randomly using a predefined distribution $\pi$. The generated batch is fed into a "discriminator + classifier" network with shared weights in all layers except the output layer. The goal of the classifier is to predict which generator generated the images, and the discriminator has the same functionality as the vanilla GAN method.

MAD-GAN (Ghosh et al., 2018) is a method which combines the classifier and discriminator functionalities into a single model.

All of the previously mentioned methods require that an extra model be trained with the original GAN or multi-generator setup to force separation of modes among the generators. Unlike these methods, AdaGAN introduces a sequential "training+adding generators" approach. This approach is computationally inefficient as shown by Hoang et al. (2017).

Although the results suggest that these methods work well, they require adding an extra step in the training process. The method presented in this contribution manages to achieve promising results without the need of such additions.

## 2.3 GAN EVALUATION METRICS

The most commonly used metrics in the GAN community to evaluate the generated images is the Inception Score (IS) as introduced in Salimans et al. (2016). It uses the pre-trained InceptionV3 model (Szegedy et al., 2014) to predict the class probabilities, the conditional probability of class prediction given the image $p(y|x)$. The marginal distribution $p(y)$ is calculated using the batch-wise average of $p(y|x)$. Finally the KL-Divergence is calculated using

$$KLD = p(y|x) * (log(p(y|x)) - log(p(y)))$$ 
(4)

This is performed over multiple batches and the final score is reported as the mean and standard deviation of all the batches.

Over the past few years, IS has started to be considered an unreliable metric for evaluation, especially for datasets that are not ImageNet (Barratt & Sharma, 2018). A better evaluation metric that has become more popular is the Frechet Inception Distance (FID) (Heusel et al., 2017). This metric uses the InceptionV3 model without the final classification layer. The latent distribution generated by the InceptionV3 network for real and fake images is compared. Specifically, the FID metric is given by

$$FID = (\mu_1 - \mu_2)^2 + Tr(C_1 + C_2 - \sqrt{C_1 * C_2})$$ 
(5)

where $\mu_1$ and $\mu_2$ are means of the latent representation of fake and real data, respectively and $C_1$ and $C_2$ are their covariance matrices. This metric was proposed as an improvement over the IS. However, it suffers from some of the same issues as the IS because of the fact that InceptionV3 is pre-trained on a 1000 class dataset of RGB images.

For our particular use case with MNIST (a grayscale dataset with only 10 classes), we used the `tensorflow-gan` library's built in method `mnist_frechet_distance` function. This function uses a pre-trained MNIST classifier, rather than the less relevant InceptionV3, to calculate the Frechet distance.

## 3 METHODS

### 3.1 DATASET

MNIST is a very well-known dataset. It contains 60,000 images (train set) and 10,000 images (test set) of handwritten digits belonging to 10 classes (numbers 0-9) of 28x28x1 pixels per image. It is widely used for proof-of-concepts of various model architectures and training protocols related to image classification or generation. This is because it is easily available through TensorFlow and PyTorch, smaller images make it easier to implement on simplistic hardware and it has well-defined modes in the dataset. This is the dataset we used for our experiments. The train set was used during the training procedures and the test set was used to evaluate our models. We resized the images to 32x32 to maintain the dimensions as a power of 2, to maintain even dimensions while using strided convolutions. We also normalized pixel values to lie between -1 and 1, as this has been shown to perform better in WGAN-GP (Gulrajani et al., 2017).

### 3.2 THE TVD LOSS COMPONENT

The Total Variation Distance (TVD) is a quantity that measures the distance between two categorical distributions and is defined as:

$$TVD(P,Q) = \frac{1}{2} \sum_{x \in \mathbf{X}} |P(x) - Q(x)| \tag{6}$$

Where $P$ and $Q$ are two categorical distribution vectors, $\mathbf{X}$ are the number of categories in the distribution and $P(x)$ and $Q(x)$ denote the probability of class $x$ in the distribution $P$ and $Q$ respectively. We assumed each class in the MNIST dataset is a mode. To enforce mode separation among individual generators, we augmented the generator loss in the original WGAN-GP paper (Gulrajani et al., 2017) with a TVD based component. We chose this as the preferred metric because it has well defined bounds ($TVD(P,Q) \in [0,1]$) and is easy to compute for categorical distributions. It is also commutative, that is $TVD(i,j) = TVD(j,i)$.
Subsequently, we defined the distance metric:

$$\Delta = \frac{1}{N} \sum_{i=1}^{n} \sum_{j=i+1}^{n} TVD(C(\tilde{\mathbf{x}}_i), C(\tilde{\mathbf{x}}_j)) \tag{7}$$

which is just the average pairwise TVD. Here, $C(\tilde{\mathbf{x}}_i)$ and $C(\tilde{\mathbf{x}}_j))$ are categorical distributions generated by a pre-trained MNIST classifier (C) on $\tilde{\mathbf{x}}_i$ and $\tilde{\mathbf{x}}_j$, the images generated by generator $i$ and $j$ respectively. The classifier is a very basic convolution neural network. Refer to Appendix A for more information on classifier training. We obtained a peak accuracy of 99% after training. $N = n(n-1)/2$ is the number of unique pairs of categorical distributions using 'n' generators. Finally, we introduced this term to the loss of each generator to get:

$$L_{G,i} = -\mathbb{E}_{\tilde{\mathbf{x}}_i \sim P_{Gi}} D(\tilde{\mathbf{x}}_i) + \gamma * (1 - \Delta) \qquad i = 1, 2, ..., n \tag{8}$$

where the first term is the original generator loss from the WGAN-GP paper. The hyperparameter $\gamma$ is the distance weight. We carried out experiments to find the optimal $\gamma$ which produces the best possible FID (refer to Section 4).

### 3.3 THE TRAINING ALGORITHM

The training algorithm remained the same for all the experiments. However, in each experiment, the effect of each newly introduced parameter (namely, $\gamma$ and $n$) was tested. The specific values used for each experiment, are specified in Section 4. All other hyperparameters are constant throughout all experiments. The batch size $m$ has to be varied to maintain divisibility by $n$ in step 4 of the algorithm. However, we restricted this value to one of two possible values $[500, 504]$, to maintain approximate similarity. Specifically, for $n = 1, 2, 4, 5, 10$, $m = 500$ and for $n = 3, 6, 7, 8, 9$, $m = 504$. We used the generator learning rate and discriminator learning rate $\alpha_g = \alpha_d = 0.0002$. We chose the number of discriminator training iterations per generator iteration, $n_d = 3$, following the recommendations of Gulrajani et al. (2017). Also following their recommendations, we used the *Adam* optimizer, setting $\beta_1 = 0.5$ and $\beta_2 = 0.9$ for all our experiments. The ∘ symbol denotes concatenation along batch dimension. $\lambda = 10$ is the gradient penalty weight as described in Eq. equation 3. The algorithm pseudocode can be found in Algorithm 1 below.

---

**Algorithm 1** Training algorithm

---

**Require:** $\beta_1$, $\beta_2$, $\alpha$, the *Adam* parameters; $\gamma$ the distance weight; $n$ the number of generators; $n_d$ the number of discriminator iterations per generator iteration; $\lambda$ the GP coefficient; $m$ the batch size; $\alpha_g$, $\alpha_d$, the learning rates; a pre-trained classifier (C).
**Require:** initial discriminator parameters $w_0$, initial generator parameters $\theta_{0,i}\forall i = 1, 2, ..., n$

1: **while** $\theta_i$ have not converged **do**
2:     **for** $t = 1, ..., n_d$ **do**
3:         Sample real data batch $\mathbf{x} \sim P_r$, latent variable batch $\mathbf{z} \sim p(\mathbf{z})$ and $\epsilon \sim U[0, 1]$.
4:         Split the latent vector batch into $n$ sub-batches $(\mathbf{z}_1, ..., \mathbf{z}_n)$
5:         $\tilde{\mathbf{x}}_i \leftarrow G_i(\mathbf{z}_i) \qquad \forall i = 1, ..., n$
6:         $\tilde{\mathbf{x}} \leftarrow \tilde{\mathbf{x}}_1 \circ \tilde{\mathbf{x}}_2 \circ ... \circ \tilde{\mathbf{x}}_n$
7:         $\hat{\mathbf{x}} \leftarrow \epsilon\mathbf{x} + (1 - \epsilon)\tilde{\mathbf{x}}$
8:         $L_d \leftarrow \frac{1}{m}\sum_{i=1}^{m}[D_w(\tilde{\mathbf{x}}) - D_w(\mathbf{x}) + \lambda(||\nabla_{\hat{\mathbf{x}}}D(\hat{\mathbf{x}})||_2 - 1)^2]$
9:         $w \leftarrow Adam(\nabla_w L_d, w, \alpha, \beta_1, \beta_2)$
10:     **end for**
11:     sample $n$ latent variable batches $\mathbf{z}_i \sim p(\mathbf{z}) \qquad \forall i = 1, 2, ..., n$
12:     $\tilde{\mathbf{x}}_i \leftarrow G_i(\mathbf{z}_i) \qquad \forall i = 1, ..., n$
13:     **if** $\gamma \neq 0$ **then**
14:         $\Delta \leftarrow \frac{1}{N}\sum_{i=1}^{n}\sum_{j=i+1}^{n}TVD(C(\tilde{\mathbf{x}}_i), C(\tilde{\mathbf{x}}_j))$
15:         $L_{g,i} \leftarrow \frac{1}{m}\sum_{k=1}^{m}[-D(\tilde{x}_i^{(k)}) + \gamma(1 - \Delta)] \qquad \forall i = 1, 2, ..., n$
16:     **else**
17:         $L_{g,i} \leftarrow \frac{1}{m}\sum_{k=1}^{m}-D(\tilde{x}_i^{(k)}) \qquad \forall i = 1, 2, ..., n$
18:     **end if**
19:     $\theta_i \leftarrow Adam(\nabla_{\theta_i}L_{g,i}, \theta_i, \alpha, \beta_1, \beta_2) \qquad \forall i = 1, 2, ..., n$
20: **end while**

---

## 4 EXPERIMENTS

For all the experiments below, we used the same model architectures for all the generators and the discriminators, the details of which can be found in Appendix B.
We performed 3 broad categories of experiments -

1. Baseline experiments as a proof of concept of our new method.
2. Optimizing for the hyperparameter $\gamma$.
3. Experiments comparing number of generators versus performance (in terms of FID).

In the baseline experiments, we ran two types of tests:

- Model size experiments
- TVD loss experiments

In the first type, we ran the original WGAN-GP procedure, with bigger generators and compared performance with 4 small generators using our method. We used two different ways to make the generators bigger - using dense layers to increase depth and using residual blocks. When using 4 small generators, we used $\gamma = 12.5$. We kept the discriminator the same throughout these experiments. All three models were trained for 70 epochs. In the second type, we ran two experiments, one with $\gamma = 0$ (termed as *batch concatenation*) and one with $\gamma = 5.0$, an arbitrary value to see the impact of the new loss component on convergence. Here we trained all models until convergence was reached.

In the second category, we fixed the number of generators $n = 5$ and the number of epochs = 70. We carried out 3 tests:

- Constant $\gamma$ throughout training at discrete values $[0, 5, 10, 11, 12, 13, 14, 15, 20]$.
- Linear decay from $\gamma = 20$ to $\gamma = 5$, so that the effective weight is equal to the average of the two ($\gamma_{eff} = 12.5$), which we found to be close to the optimal for the setup.
- Sudden death, where $\gamma = 20$ for the first 20 epochs of training and then it is equal to 0 for the remainder.

In the final category, we swept the number of generators $n = [1, 2, ..., 10]$ to see how it affects the performance. In this category of experiments, the constant parameter $\gamma = 12.2$ was chosen after the optimizations performed in the second category of experiments. Here, we trained all models until convergence was reached.

## 5 RESULTS AND DISCUSSION

If we run our experiments, with $n = 1$, we can also evaluate the state-of-the-art FID for a vanilla WGAN-GP. We performed a session of training under these conditions and as can be observed in the Fig. 4, the current state-of-the-art WGAN-GP gives us an $FID \sim 4.25$. In all the future discussions, we will compare the obtained values to this one.

### 5.1 BASELINE EXPERIMENTS

#### 5.1.1 MODEL SIZE EXPERIMENTS

In all experiments the discriminator was identical and had $4,306,177$ parameters. In the first experiment, we trained 4 generators using the TVD loss method ($\gamma = 12.5$). Each generator model was the same as presented in B and had $910,660$ parameters for a total of $3,642,640$ parameters. Running these 4 small generators resulted in a final average $FID = 2.6$. In the second experiment, we added an extra *dense* layer to the generator model, resulting in $69,657,924$ parameters ( $76.5$ times the original generator). After training using the WGAN-GP protocol, it converged to an average $FID = 5.56$. In the third experiment, we replaced the generator with a much deeper network that uses residual blocks. The total number of parameters in the generator is $3,882,049$ ( $4.3$ times the original generator). This model did not converge because the discriminator was easily overpowered. The average $FID = 23.42$. This model converges if we add more parameters to the discriminator. The models used in the latter two experiments are also available in B. From the above results, it is clear that two or more competing generators exceed the performance of bigger and more complex ones. Our method can be efficiently parallelized due to the architecture.

#### 5.1.2 TVD LOSS EXPERIMENTS

We ran both of the two generators experiments until convergence 3 times, to test robustness. The convergence graphs for one training session are presented in Fig. 1. Fig. 1b shows the convergence graph for batch concatenation (b.c.) + TVD and Fig. 1a shows just batch concatenation (b.c.) based generator loss. We see from the graphs that both of the generators are converging at a similar pace (the lines overlap with each other), under both loss functions. The discriminator loss going to 0 indicates that it predicts negative and positive values with almost equal probability (meaning the system has converged). The network managed to converge all 3 times for both loss functions ($\gamma = 0$ and $\gamma = 5$).

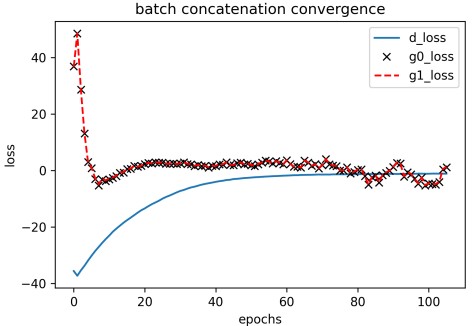

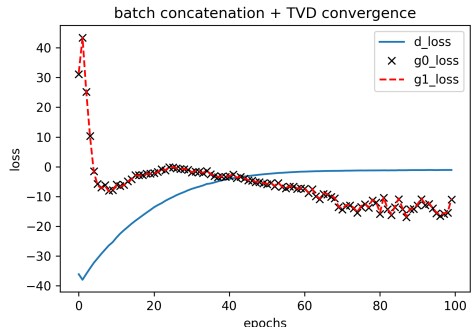

(a) Convergence graphs for batch concatenation only

(b) Convergence graphs for batch concatenation augmented with TVD

Figure 1: 2 Generator convergence graphs

After convergence, we measured **3 metrics**: FID, calculated according to Eq. equation 5; $TVD(C(G_1(\mathbf{z}_1)), C(G_2(\mathbf{z}_2)))$ and; average class probabilities (softmax) for each generators' generations, as predicted by C. These metrics are presented in Tables 1 and 2.

Table 1: FID and TVD metric comparison

| Loss | FID | TVD |
|------|-----|-----|
| b.c. only | 4.738 | 0.9832 |
| TVD + b.c. | 3.602 | 0.9965 |

Table 2: Average class probabilities

| b.c. only | 0 | 1 | 2 | 3 | 4 | 5 | 6 | 7 | 8 | 9 |
|-----------|---|---|---|---|---|---|---|---|---|---|
| generator 0 | 00.00 | **27.23** | **13.23** | 00.63 | **18.91** | 00.00 | 00.00 | **22.46** | 00.32 | **17.23** |
| generator 1 | **19.15** | 00.01 | 08.94 | **19.75** | 00.06 | **17.19** | **18.35** | 00.29 | **14.93** | 01.31 |
| **TVD+b.c.** | 0 | 1 | 2 | 3 | 4 | 5 | 6 | 7 | 8 | 9 |
| generator 0 | **18.98** | **21.77** | 00.47 | 00.03 | **18.96** | 00.00 | 00.13 | **19.47** | 00.49 | **19.69** |
| generator 1 | 00.51 | 00.04 | **21.25** | **19.79** | 00.16 | **17.90** | **23.48** | 00.01 | **16.82** | 00.05 |

We see in Table 2 that the two generators specialize in generating disjoint subsets of classes. The actual samples generated by each generator can be seen in Fig.2. We see that they are indeed from separate classes for different generators and of very high quality. We can also observe (Table 1) how introducing a small non-zero constant $\gamma = 5$ drastically improves the FID as well as the TVD.

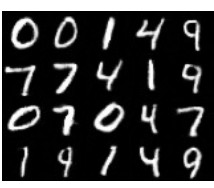 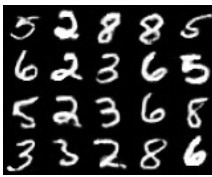 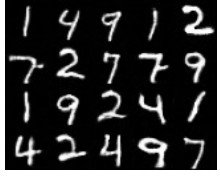 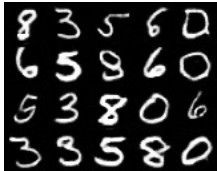

(a) Samples generated by gen0 (**tvd+b.c.**)

(b) Samples generated by gen1 (**tvd+b.c.**)

(c) Samples generated by gen0 (**b.c. only**)

(d) Samples generated by gen1 (**b.c. only**)

Figure 2: Samples generated by both the methods

## 5.2 TVD LOSS WEIGHT EXPERIMENTS

### 5.2.1 DISCRETE CONSTANT

In this experiment we swept gamma across the following values - $[0, 5, 10, 11, 12, 13, 14, 15, 20]$ and kept it constant throughout the 70 epochs of training. We fixed the number of generators to 5. We evaluated each time using the FID metric. The plot of the parametric sweep is seen in Figure 3. We can see that there is an optimum value for the TVD loss weight (approximately $\gamma = 12.2$).

### 5.2.2 LINEAR DECAY

In this experiment, we again fixed the number of generators to 5 and the number of epochs to 70. We linearly decayed the parameter $\gamma$ from 20 to 5, giving us an effective $\gamma_{eff} = 12.5$. We evaluated the FID and TVD after training is complete and received $FID = 3.5379$ and $TVD = 0.9999$. This method clearly gives an improvement over the state-of-the-art FID as well as optimal mode separation.

### 5.2.3 SUDDEN DEATH

Keeping the other elements of the setup exactly the same, we now keep $\gamma = 20$ for the first 20 epochs of training and then $\gamma = 0$ for the remaining 50. We end up with $FID = 2.6685$ and

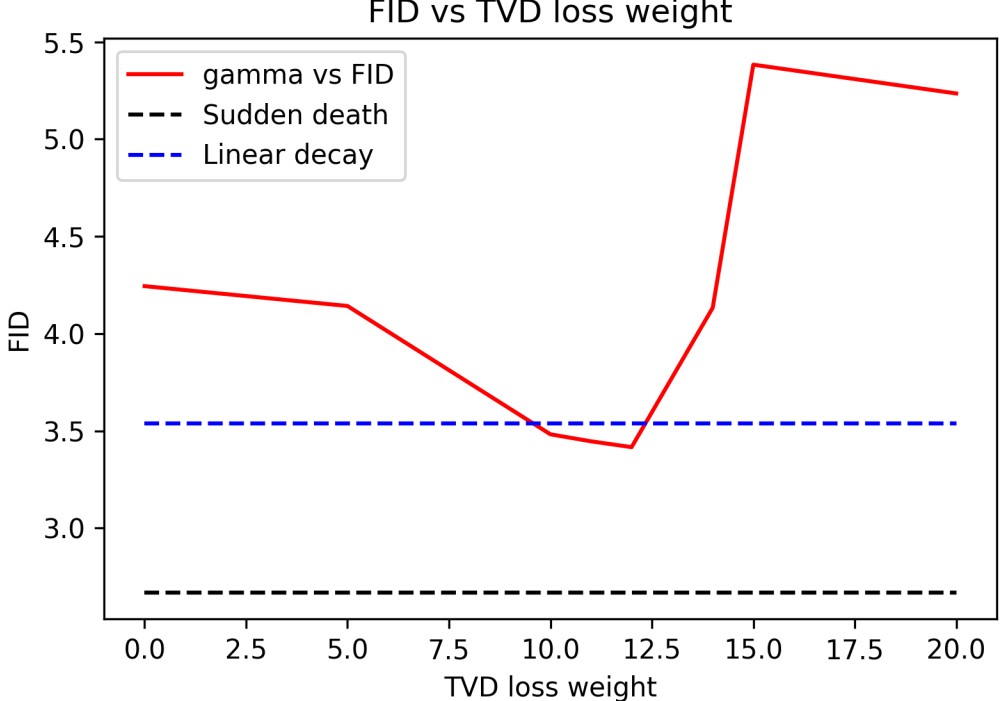

Figure 3: FID vs $\gamma$ sweep

$TVD = 0.9998$, which again is an improvement over the state-of-the-art FID along with great mode separation.

This result is interesting as it suggests that one does not need to enforce mode separation (or compute the pairwise TVD) for all of the training epochs. Once the modes have sufficiently separated, we can remove the TVD loss component and the generators will keep generating separate modes and in a much better quality than current state-of-the-art.

### 5.3 NUMBER OF GENERATORS SWEEP

From the above experiments we found an optimal value for $\gamma = 12.2$ which we fixed and kept constant for this experiment. We changed $n$ from 1 to 10 (1 representing current state-of-the-art). We trained each setup until convergence or 250 epochs, whichever came first. We set the early stopping criterion as: *less than 0.01 change in the objective for 5 consecutive epochs*. We decided to do this because when we tried running for a fixed number of epochs, we realized that for $n = 3, 6, 7, 9$ the generators did not converge in the same number of epochs. Here we notice that the mode separation problem is much more difficult because of the non-divisibility of 10 (the number of modes) by n, which results in one or more modes of the dataset being *assigned* to a different generator during different training epochs. As we can also see in Fig. 4, the FID values for these specific values of $n$ are much higher, denoting a much worse performance. The FID for $n = 4$ and $n = 5$ is the lowest, showing us that having more generators does not imply better performance and this is another parameter that can be optimized.

## 6 CONCLUSIONS

We introduced a novel approach in training WGANs in a collaborative manner which we call ComMutE-GANs. This approach is inspired by the principles of co-creativity. Our approach gives improved generation performance under space constraints as we saw in section 5.1.1. This method also manages to improve overall stability, robustness and performance of GANs. We also introduced

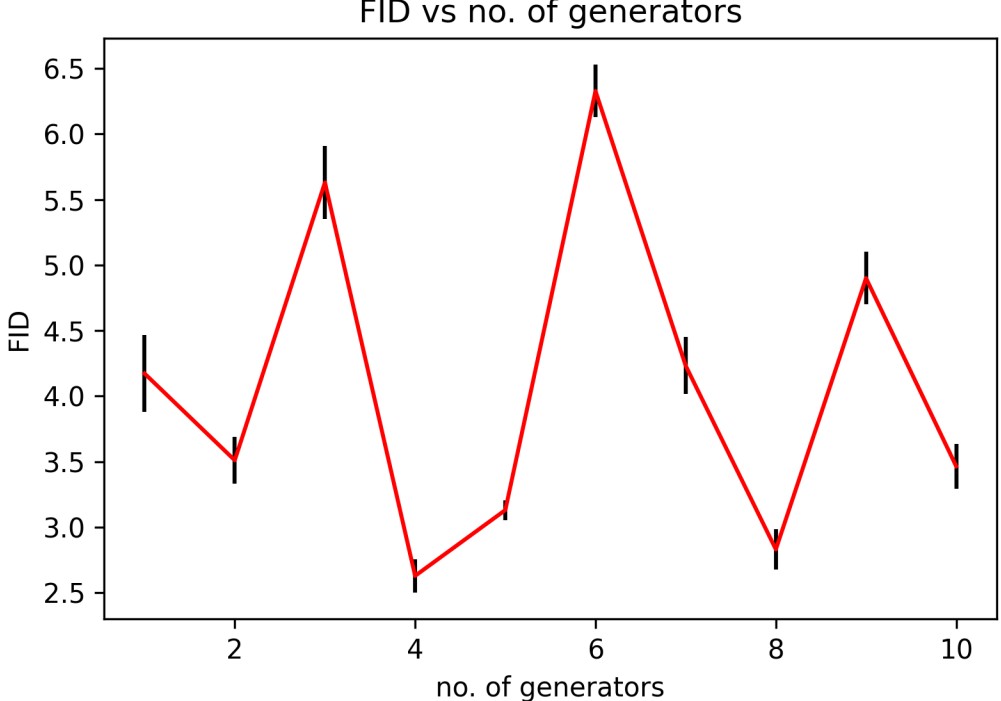

Figure 4: FID vs no. of generators

a new loss component that can be added to the generator loss during training. We observed that even adding this loss for a short amount of time in the initial training can drastically improve performance. We can also tune the weight of this loss to fit the number of generators we plan to use.

A major disadvantage of the current method is the fact that we need a pre-trained classifier to calculate a loss during training, which is not efficient. Majority of the future work will involve finding better losses to enforce mode separation without the use of a classifier. The eventual goal of this research is to use multiple simple generator models to generate multi-modal data rather than a single large, complex network. This research is a also contribution towards modelling co-creativity in humans. Study of this model can help us better understand and implement creativity promoting interactions.

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

## A  APPENDIX - CLASSIFIER TRAINING

We present specific information about the classifier in this section. The network architecture Fig. 5 and convergence graphs Fig. 6 are presented below. Each conv_block contains a *convolutional layer* with a 5x5 filter and number of filters going from 64 to 512 in powers of 2. Each *convolutional layer* is followed by *BatchNormalization*, *ReLU* and *Dropout* with $drop\_prob = 0.3$. The MLP block contains 2 layers of 128 neurons each followed by a 10 neuron output layer. As observed in 6b, the peak accuracy of our classifier was $\sim 99\%$.

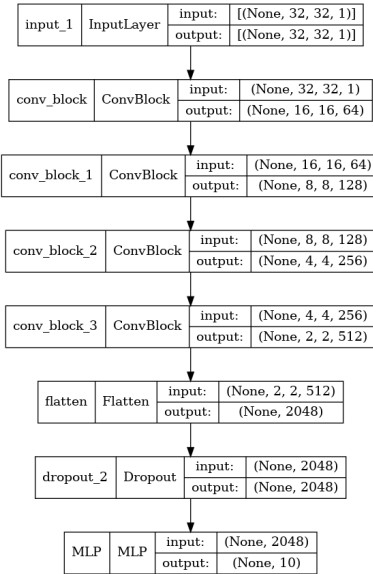

Figure 5: Classifier model

## B  APPENDIX - GENERATOR AND DISCRIMINATOR MODELS

We implemented very simple *Generator* and *Discriminator*. The exact models are presented in Fig. 7a and Fig. 7b. Each conv_block is exactly the same as the one implemented for the *Classifier*, except the use of *Layer normalization* instead of *Batch normalization*, as recommended in Arjovsky et al. (2017). Each upsample_block consists of a *2D Upsampling layer* with a factor of 2x2 followed by a *convolutional layer* with filter size 3x3 and stride 1x1. These operations are followed by *LeakyReLU* activation with $\alpha = 0.2$. *Batch normalization* is applied after the activation. The final upsample_block also uses *dropout* with $drop\_prob = 0.3$. The final upsample_block uses the *tanh* activation function to squash the values between -1 and 1.

In the model size experiments (Section 5.1.1) we extend the generator model by one extra dense layer of size $16,384$ to the structure above. The final model can be seen in fig 8.

We also replaced the above mentioned generator with a generator that uses residual blocks in the generation. The full architecture can be seen in fig 9.

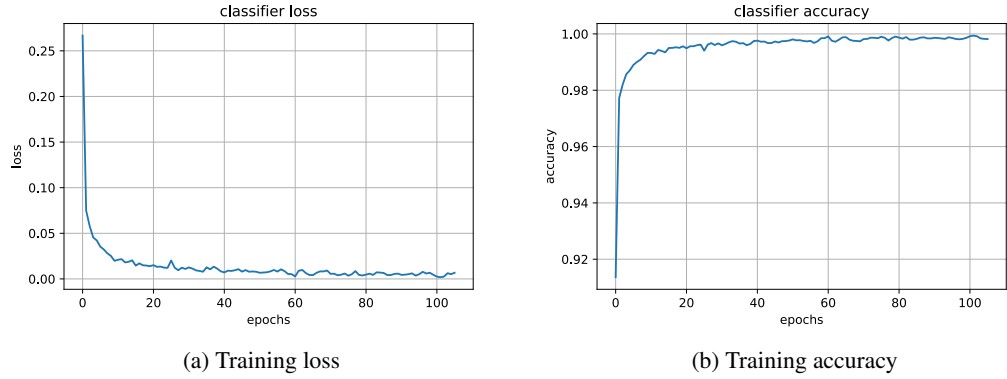

(a) Training loss        (b) Training accuracy

Figure 6: Classifier convergence graphs

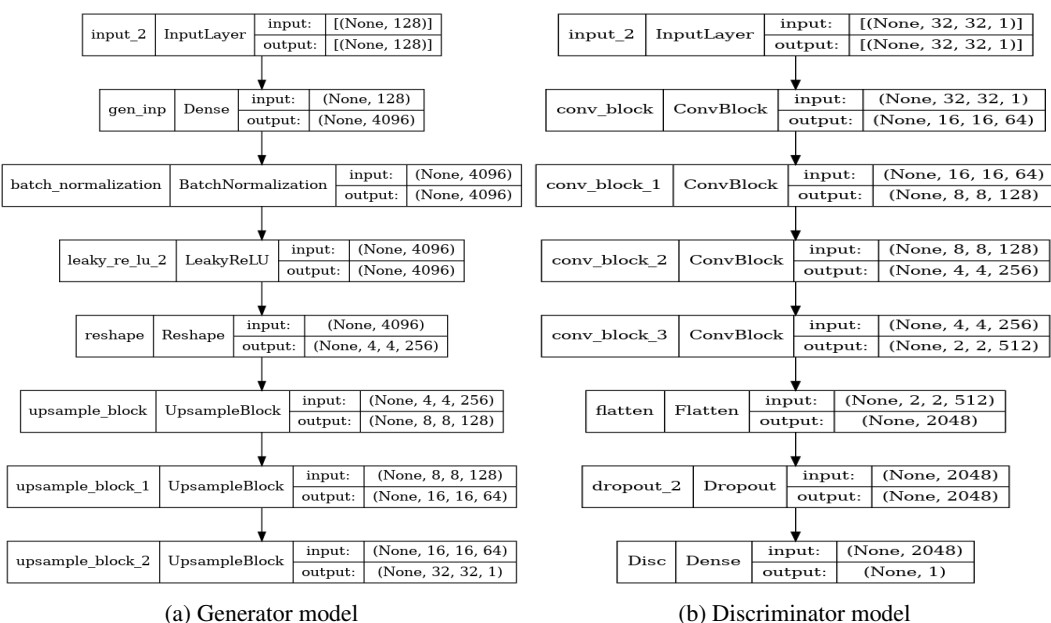

(a) Generator model        (b) Discriminator model

Figure 7: Models

## C  APPENDIX - OTHER EXPERIMENTS

Setting $\gamma = 0$ and $n = 4$, we ran the setup until convergence. This method is an extended version of the batch concatenation (b.c.) method as described in 4, to four generators instead of two. The convergence graphs show that all 4 generators converge at the same pace (the lines overlap) and the discriminator loss converges to 0 (Fig. 10). Our method improves the state-of-the-art with an $FID = 2.9439$ implying that high quality images are generated that match the dataset distribution. Table 3 and Fig. 11 show the average class probabilities and correspondingly generated samples. This method of training results in an average pairwise $TVD = 0.9633$.

In Table 3, the values in bold represent which generator gives the maximum probability over that class. The number 2 is generated by generator 1, 2 and 3 and the "workload" required for this digit is being distributed among the three of them. We can see that for $\gamma = 0$, the TVD metric goes down as expected, but we can still obtain a comparable FID to what we can achieve using the TVD loss component.

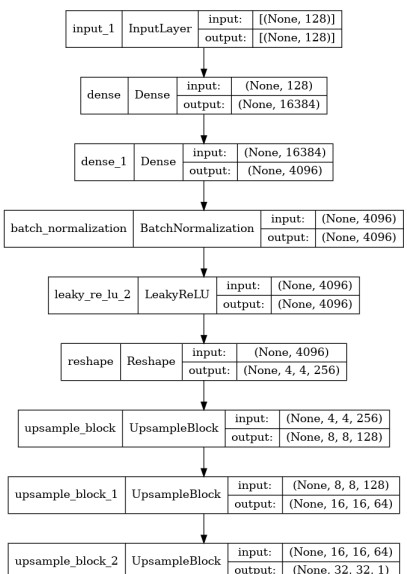

Figure 8: Deep generator model

Table 3: Average class probabilities($\gamma = 0$, $n = 4$)

| Generator number | 0 | 1 | 2 | 3 | 4 | 5 | 6 | 7 | 8 | 9 |
|---|---|---|---|---|---|---|---|---|---|---|
| generator 0 | 00.02 | 00.15 | 00.04 | 13.00 | 06.91 | 04.92 | **36.91** | 00.11 | **33.14** | 04.79 |
| generator 1 | **38.99** | 00.00 | 11.32 | **22.84** | 00.01 | **24.13** | 00.03 | 00.00 | 02.33 | 00.33 |
| generator 2 | 00.15 | **45.49** | 15.88 | 00.14 | **31.99** | 00.68 | 02.65 | 00.86 | 00.84 | 01.31 |
| generator 3 | 00.36 | 00.29 | **22.05** | 02.46 | 00.72 | 00.95 | 00.00 | **41.96** | 01.94 | **29.24** |

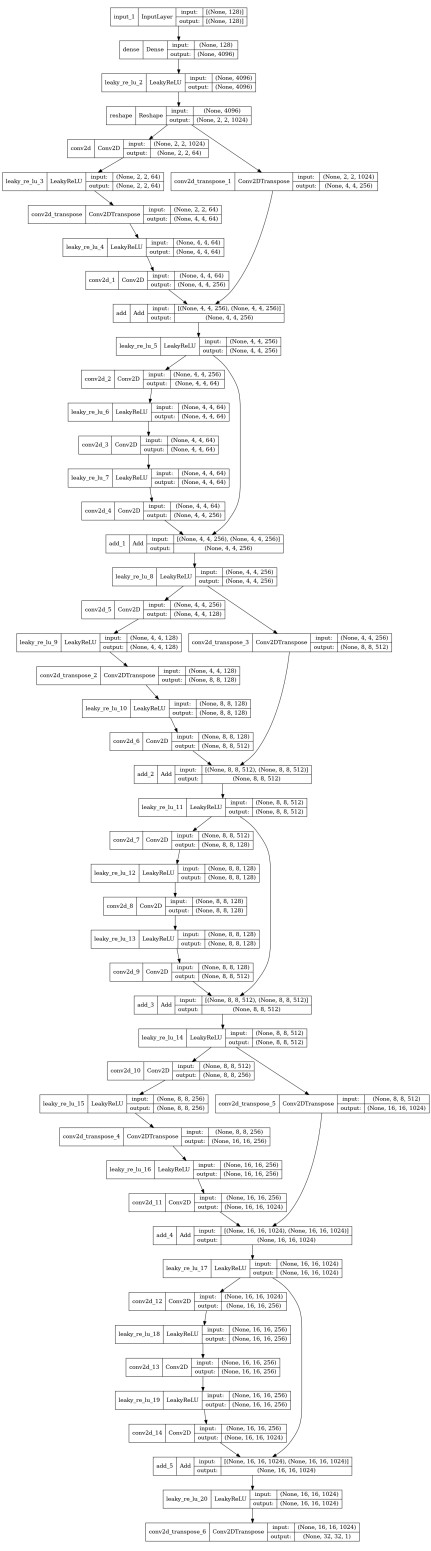

Figure 9: Deep generator model

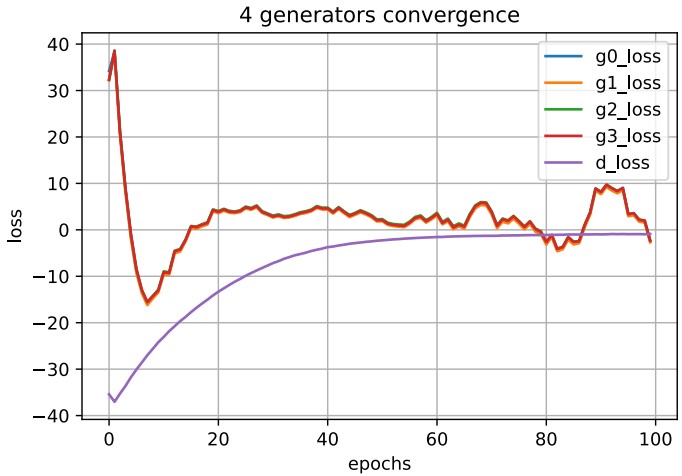

Figure 10: Convergence graph for 4 generators

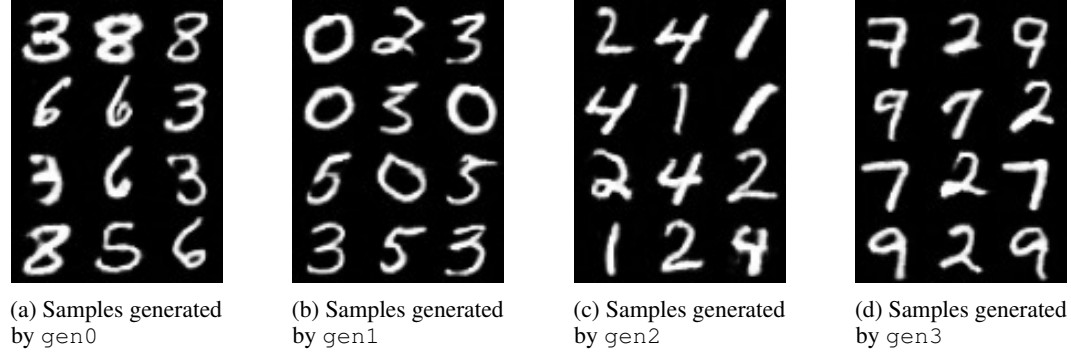

(a) Samples generated by `gen0`

(b) Samples generated by `gen1`

(c) Samples generated by `gen2`

(d) Samples generated by `gen3`

Figure 11: Samples generated by the 4 generators

