# OpenReview forum: "CoGANs: Collaborative Generative Adversarial Networks"
_ICLR.cc/2023/Conference — Submitted to ICLR 2023_

### Official Review · Reviewer_YBt6 · 2022-10-15

**Confidence:** 5
**Correctness:** 2
**Technical Novelty And Significance:** 2
**Empirical Novelty And Significance:** 2
**Recommendation:** 3

**Clarity, Quality, Novelty And Reproducibility:**

Clarity is good, easy to follow. Unusual start for a technical paper to talk about 7Cs. I'm not opposed to this per se, but I would have liked to read a bit more why the problem exists in practice. Having worked a lot with GANs, I'm sure there is no need for multiple generators in MNIST, even though the authors manage to find support for that in their context. A single (StyleGAN-XL) generator can deal even with ImageNet's 1000 classes after all.

The background of GANs is also a bit too verbose for my taste, and may confuse things a little bit. The best current GANs continue to use the (non-saturating) loss for Goodfellow's original paper. WGAN was a very interesting step, but its loss was never used in SOTA methods -- but its suggestion of a Lipschitz regularizer stuck (in the form of R1).

TVD has some novelty and seems quite interesting.

**Strength And Weaknesses:**

Pros:
- TVD loss may be a novel idea, seems promising.
- Clear exposition.

Cons:
- The existence of the problem is not defended adequately.
- MNIST is not enough. I don't believe it actually needs multiple generators.
- SOTA GANs not used, conclusions may be incorrect.
- Low information density for an ICLR paper.

**Summary Of The Paper:**

This paper studies the possibility of using multiple generators in a GAN setting. The idea is to let each generator specialize to a certain set of modes -- this "assignment" is currently assisted by a classifier, and encouraged via a novel total variation distance (TVD) loss. This idea may well have some merit, although the use of a classifier seems quite limiting. Unfortunately, the paper has pretty low information density and reads like a course work rather than an ICLR paper. The only dataset is MNIST, which is woefully inadequate in 2022. Realistically the multimodality problem should not exist in MNIST, as even in ImageNet StyleGAN-XL has SOTA results implying that modern GANs scale pretty far. So, while I agree that the problem still exists in the limit of highly diverse datasets, it surely should not exist in MNIST (using a StyleGAN family, for example).

**Summary Of The Review:**

Basically the scientific novelty, tested datasets and thus the breadth of conclusions, as well as the information density are all clearly below ICLR standards. A resubmission to a workshop might be viable option.

---

> ### Author Response · Authors · 2022-11-17
> **Thank you for your helpful comments**
>
> We first wish to thank the reviewer for their constructive comments.
>
> The existence of the problem is not defended adequately.
> >> We are exploring a new direction in an efficient use of parameters for the GAN architecture. We show an efficient implementation of adding more parameters to several competing generators. We have added a new analysis showing this point (Section 5.1.1), where we compared a larger single generator to several smaller generators using our method. We show that adding more parameters using our methods (multiple generators) is more efficient and gives better results, than adding the same (or even more) parameters to a single generator.
>
> MNIST is not enough. I don't believe it actually needs multiple generators.
> >> Our approach comes from a different direction. Given a specific (small) amount of parameters, what is the best utilization of these parameters? Hence, even if a large single generator can solve MNIST, we show that with the same (or even less) parameters, we improve the results (Section 5.1.1)
>
> SOTA GANs not used, conclusions may be incorrect.
> >> We changed the conclusions, from beating SOTA-GANs to a more efficient use of memory in the form of multiple generators.
>
> Low information density for an ICLR paper.
> >> We reduced the Introduction, and the basic GANs architectures. We added more explanations regarding our method, as well as new results (Section 5.1.1).
>
> Unusual start for a technical paper to talk about 7Cs. I'm not opposed to this per se, but I would have liked to read a bit more why the problem exists in practice.
> >> We added several sentences to the Introduction to better frame our approach, namely, the efficient use of memory via multiple small generators.

---

### Official Review · Reviewer_f1XE · 2022-10-26

**Confidence:** 4
**Clarity, Quality, Novelty And Reproducibility:** The clarity and novelty of this paper…
**Correctness:** 2
**Technical Novelty And Significance:** 1
**Empirical Novelty And Significance:** Not applicable
**Recommendation:** 3

**Strength And Weaknesses:**

1) The novelty of the proposed method is very limited. It simply adds additional TVD loss on the existing WGAN-GP. And it is not clear why adding such TVD loss to the WGAN-GP can optimize the workload of the individual generators.

2) The organization of this paper is poor. The paper spends too much space on introducing existing models and works, while the introduction of the proposed model is too concise.

3) The experiments are only conducted on the simple MNIST datasets. The results are not convincing at all.

**Summary Of The Paper:**

The paper presents collaborative GAN, which aims to better balance the workload between individual generators. Specifically, a pre-trained classifier is introduced, and the training strategy is modified by adding a total variation distance based on WGAN-GP. Some experiments were conducted on MNIST dataset.


**Summary Of The Review:**

The novelty and experiments of this paper are in doubt.

---

> ### Author Response · Authors · 2022-11-17
> **Thank you for your helpful comments**
>
> We first wish to thank the reviewer for their constructive comments.
>
> The novelty of the proposed method is very limited. It simply adds additional TVD loss on the existing WGAN-GP. And it is not clear why adding such TVD loss to the WGAN-GP can optimize the workload of the individual generators.
> >> We have added a new analysis showing this point (Section 5.1.1), where we compared a larger single generator to several smaller generators using our method. We show that adding more parameters using our methods (multiple generators) is more efficient and gives better results, than adding the same (or even more) parameters to a single generator.
>
> The organization of this paper is poor. The paper spends too much space on introducing existing models and works, while the introduction of the proposed model is too concise.
> >> We reduced the Introduction, and the basic GANs architectures. We added more explanations regarding our method, as well as new results (Section 5.1.1)

---

### Official Review · Reviewer_WzHB · 2022-10-26

**Confidence:** 5
**Correctness:** 2
**Technical Novelty And Significance:** 1
**Empirical Novelty And Significance:** 1
**Recommendation:** 1

**Clarity, Quality, Novelty And Reproducibility:**

Clear writing, but vague formulations for the equations and the algorithm.

Zero practical usage and zero new theoretical thoughts.

Easy to reproduce.



**Strength And Weaknesses:**

*************************
Strengths
*************************
+ Clear writing, except for the equations and the algorithm.
+ Easy to reproduce.

*************************
Weaknesses
*************************

- The name “CoGAN” has been used in [r1].

- The motivation is meaningless.
  - It is only applicable to data with finite discrete modes. And the complexity (number of generators) is proportional to the number of modes. This is one of the major reasons their experiments only involve the toy-like MNIST dataset, and impossible for Stacked MNIST [r2] with 1000 modes nor any other natural images like FFHQ human faces.
  - The reviewer suggests authors stop thinking along this direction, which is deviated from the standard prototypes of research, has zero practical usage and zero new theoretical thoughts.

- Technical details are vague and inaccurate.
  - In Eq. 6, what are the definitions of P and Q? What is the distance metric used to measure |P(x) - Q(x)|? It has a summation over a set of x and, for each summation term, P and Q take the same x. This differs a lot from the formulation of delta, where x_i and x_j are different single samples not a set. Please correct and articulate your formulations.
  - In Algorithm 1, k is not reflected in the summation terms when calculating L_g.

- Experiments are far from convincing.
  - The dataset includes only the toy-like MNIST dataset. Actually the method itself cannot be applicable to other realistic datasets.
  - The WGAN-GP backbone is out-of-date. Experiment with multiple backbones including the recent SOTAs like StyleGAN-XL [r3]. Please also discuss the recent progress in GANs, to convince readers that the authors are knowledgeably ready to research in the GAN regime.

[r1] Liu, Ming-Yu, and Oncel Tuzel. "Coupled generative adversarial networks." NeurIPS 2016.
[r2] Metz, Luke, et al. "Unrolled generative adversarial networks." NeurIPS 2017.
[r3] Sauer, Axel, Katja Schwarz, and Andreas Geiger. "Stylegan-xl: Scaling stylegan to large diverse datasets." SIGGRAPH 2022.

**Summary Of The Paper:**

The authors target to learn a set of collaborative generators and a single discriminator in the generative adversarial network (GAN) framework. They are motivated by the belief that collaboration among multiple skill-specific generators can reach general and complex performance. They prove the concept on the MNIST dataset with the help of Total Variation Distance (TVD).

**Summary Of The Review:**

See above.

---

> ### Author Response · Authors · 2022-11-17
> **Thank you for your honest and constructive inputs**
>
> We first wish to thank the reviewer for their constructive comments. Below we address each highlighted issue
>
> The name “CoGAN” has been used in [r1].
> >> We changed the name of our architecture to Competing Multiple Efficient GANs (ComMutE-GANs)
>
> It is only applicable to data with finite discrete modes.
> >> This is correct, yet there are many interesting problems involving GANs with discrete modes.
>
> And the complexity (number of generators) is proportional to the number of modes. This is one of the major reasons their experiments only involve the toy-like MNIST dataset, and impossible for Stacked MNIST [r2] with 1000 modes nor any other natural images like FFHQ human faces.
> >> The complexity of our proposed algorithm is proportional to the number of generators, not modes. As we have shown in the MNIST example, even 2 generators improve the results (although there are 10 modes). Hence, this is not a true limitation of our work. For the rebuttal we did not manage to run a more complex problem, but for the camera-ready deadline we will have the CIFAR-100 dataset results.
>
> In Eq. 6, what are the definitions of P and Q? What is the distance metric used to measure |P(x) - Q(x)|? It has a summation over a set of x and, for each summation term, P and Q take the same x. This differs a lot from the formulation of delta, where x_i and x_j are different single samples not a set. Please correct and articulate your formulations.
> >> We better define each notation. P and Q denote two categorical distributions. Later on, the loss is extended to a “pairwise TVD”, where
>
> C(x_i) and C(x_j) are the categorical distributions corresponding to generator i and j respectively.
> In Algorithm 1, k is not reflected in the summation terms when calculating L_g.
> >> We fixed this typo.
>
> The dataset includes only the toy-like MNIST dataset. Actually the method itself cannot be applicable to other realistic datasets.
> >> Our method has no limitations on running on larger or more complex data sets. The only limitation is that there are discrete modes (which most GANs have).
>
> The WGAN-GP backbone is out-of-date. Experiment with multiple backbones including the recent SOTAs like StyleGAN-XL [r3].
> >> Modern GANs such as StyleGAN-XL that provide state-of-the-art performance on large-scale image synthesis have larger models and more complex training procedures. Our method explores another direction - more instances of compact generators competing with each other. We have added a new analysis showing this point (Section 5.1.1), where we compared a larger single generator to several smaller generators using our method. We show that adding more parameters using our methods (multiple generators) is more efficient and gives better results, than adding the same (or even more) parameters to a single generator.
>
> Please also discuss the recent progress in GANs, to convince readers that the authors are knowledgeably ready to research in the GAN regime.
> >> We have added another paragraph to the Related Works section with more recent GANs, which we are of-course familiar with.

---

### Decision · Program_Chairs · 2023-01-20

**Decision:**

Reject

**Justification For Why Not Higher Score:**

N/A

**Justification For Why Not Lower Score:**

N/A

**Metareview: Summary, Strengths And Weaknesses:**

The paper presents collaborative GAN to better balance the workload between individual generators. The paper is easy to follow. But the motivation is very weak and there are some mistakes on the technical details. The novelty is also very limited. Thus, I recommend reject.